# TonEBP Haploinsufficiency Attenuates Microglial Activation and Memory Deficits in Middle-Aged and Amyloid β Oligomer-Treated Mice

**DOI:** 10.3390/cells12222612

**Published:** 2023-11-12

**Authors:** Jong Youl Lee, Eun Ae Jeong, Jaewoong Lee, Hyun Joo Shin, So Jeong Lee, Hyeong Seok An, Kyung Eun Kim, Won-Ho Kim, Yong Chul Bae, Heeyoung Kang, Gu Seob Roh

**Affiliations:** 1Department of Anatomy and Convergence Medical Science, College of Medicine, Institute of Medical Science, Gyeongsang National University, Jinju 52727, Republic of Korea; jyv7874v@naver.com (J.Y.L.); jeasky44@naver.com (E.A.J.); woongs1111@gmail.com (J.L.); k4900@hanmail.net (H.J.S.); thwjd5411@naver.com (S.J.L.); gudtjr5287@hanmail.net (H.S.A.); kke-jws@hanmail.net (K.E.K.); 2Division of Cardiovascular Disease Research, Department of Chronic Disease Convergence Research, Korea National Institute of Health, Cheongju 28159, Republic of Korea; jhkwh@nih.go.kr; 3Department of Anatomy and Neurobiology, School of Dentistry, Kyungpook National University, Daegu 41944, Republic of Korea; ycbae@knu.ac.kr; 4Department of Neurology, College of Medicine, Gyeongsang National University Hospital, Gyeongsang National University, Jinju 52727, Republic of Korea; miranda75@naver.com

**Keywords:** TonEBP, microglia, hippocampus, aging, amyloid beta

## Abstract

Age-related microglial activation is associated with cognitive impairment. Tonicity-responsive enhancer-binding protein (TonEBP) is a critical mediator of microglial activation in response to neuroinflammation. However, the precise role of TonEBP in the middle-aged brain is not yet known. We used TonEBP haploinsufficient mice to investigate the role of TonEBP in middle-aged or amyloid β oligomer (AβO)–injected brains and examined the effect of TonEBP knockdown on AβO-treated BV2 microglial cells. Consistent with an increase in microglial activation with aging, hippocampal TonEBP expression levels were increased in middle-aged (12-month-old) and old (24-month-old) mice compared with young (6-month-old) mice. Middle-aged TonEBP haploinsufficient mice showed reduced microglial activation and fewer memory deficits than wild-type mice. Electron microscopy revealed that synaptic pruning by microglial processes was reduced by TonEBP haploinsufficiency. TonEBP haploinsufficiency also reduced dendritic spine loss and improved memory deficits in AβO-treated mice. Furthermore, TonEBP knockdown attenuated migration and phagocytosis in AβO-treated BV2 cells. These findings suggest that TonEBP plays important roles in age-related microglial activation and memory deficits.

## 1. Introduction

Aging-related memory decline is becoming a world health issue as the human lifespan increases [1]. According to recent studies, the major risk factors of cognitive impairment, independent of amyloid deposition, are the shrinkage of neuronal synapses, chronic low-grade inflammation of glial cells, and the lack of microglial phagocytosis against neurodegenerative alterations [2,3]. Microglia are thus strongly implicated in aging-related cognitive impairment.

Microglia, as the resident macrophages of the brain, exert several functions, including chemotaxis, phagocytosis, and the production of proinflammatory cytokines against a diverse array of insults [4]. In Alzheimer’s disease (AD), activated microglia are recruited to extracellular amyloid beta (Aβ) plaques, where they release proinflammatory cytokines and cause synaptic damage [5,6]. Activated microglia also contact degenerative neurons and remove their synaptic processes, a phenomenon termed “synaptic stripping”, by engulfing synapses through phagocytosis [7]. Thus, microglial activation and phagocytosis may play critical roles in neuroinflammation during aging-related neurodegeneration.

Tonicity-responsive enhancer-binding protein (TonEBP) is a transcriptional factor of the Rel family, which includes nuclear factor-kappa B (NF-κB) [8]. TonEBP was initially found to respond to hypertonicity by inducing the expression of osmoprotective genes and promoting inflammation in renal injury [9]. More recently, tonicity-independent stimuli, such as inflammation, excitotoxicity, and hyperglycemia, were also found to increase hippocampal TonEBP expression [10,11,12,13,14]. Although TonEBP has been known to play a crucial regulatory role in the lipopolysaccharide (LPS)-induced acceleration of proinflammatory gene expression in microglia [11,13], its role in age-related microglial activation is not yet fully understood.

In this study, we investigated the roles of TonEBP in microglial activation and cognitive impairment in an aged mouse model. TonEBP expression increased in the mouse hippocampus with age. However, 12-month-old TonEBP haploinsufficient mice had reduced microglial activation and attenuated memory deficits. Furthermore, we evaluated the effects of TonEBP haploinsufficiency on neuroinflammation and dendritic spine loss in Aβ oligomer (AβO)-treated mice. Using AβO-treated BV2 microglial cells, we examined the effects of TonEBP deletion on microglial migration and phagocytosis. The results of this study suggest that TonEBP may play important roles in aging-related microglial activation and memory deficits.

## 2. Materials and Methods

### 2.1. Animals

TonEBP heterozygote mice were obtained from Dr. Kwon (Ulsan National Institute of Science and Technology, Ulsan, Republic of Korea). TonEBP haploinsufficient (+/−) mice were backcrossed with C57BL/6J mice (The Central Laboratory Animal Inc. Seoul, Republic of Korea) to produce TonEBP (+/−) mice. TonEBP (+/−) mice were used to assess the function of TonEBP in this study as the homozygous deletion of TonEBP causes a notable atrophy of the renal medulla [15]. In the aging model, male wild-type (WT) mice (*n* = 5–10) were fed a normal standard diet for 6, 12, or 24 months. Mice were kept with a 12 h light/dark cycle.

### 2.2. AβO-Treated Mouse Model

To treat the mouse model with AβO, we prepared it as previously described [16]. At 28 weeks of age, mice were anesthetized with 40 mg/kg Zoletil (Virbac Laboratories, Carros, France) and 5 mg/kg Rompun (Bayer Korea, Seoul, Republic of Korea) and were immobilized in a fully motorized stereotaxic instrument (Neurostar, Animalab, Dąbrowskiego, Poznań, Poland) and then received one injection of AβO (1 mg/mL) into the hippocampal CA1 region in the right hemisphere using a 10 μL Hamilton syringe. The stereotaxic coordinates of the injection were referenced from bregma according to Paxinos and Franklin’s Mouse Brain atlas (AP = −2.0 mm; ML = +1.3 mm; DV = −1.6 mm). A 2 μL solution of AβO (*n* = 13) or sterile saline (CTL, *n* = 13) in dimethyl sulfoxide (DMSO) was injected over 10 min. AβO-treated mice were sacrificed 1 month after AβO injection.

### 2.3. Rapid Golgi Staining

After anesthesia, brains (*n* = 3) were taken from mouse skulls as rapidly as possible and quickly washed in distilled water. Golgi staining was performed according to the manufacturer’s protocol using an FD Fast Golgi Stain Kit (FD Neuro Technologies, Inc., Ellicott, MD, USA). Coronal sections (150 μm) of Golgi-stained brains were cut with a vibratome (Leica, VT1200S, Freiburg, Germany). The slices were then placed on microscope slides, and sections were examined using a BX51 light microscope (Olympus, Tokyo, Japan). Nine fields (20 × 20 μm^2^) were randomly selected for counting the basal and apical spines in the hippocampal CA1 region.

### 2.4. Tissue Preparation

After anesthesia, mice (*n* = 3) were perfused with 4% paraformaldehyde solution in 0.1 M PBS through the left ventricle. The brains were fixed for 12 h at 4 °C, and then they were gradually immersed in 0.1 M PBS containing 15%, 20%, and then 30% sucrose at 4 °C. Frozen brains were cut into 30 μm coronal sections.

### 2.5. ProteoStat Staining Assay

To assay the aggregates and aggresomes in mouse hippocampi, we utilized a ProteoStat^®^ aggresome detection kit (Enzo Life Sciences Inc., Farmingdale, NY, USA) following the manufacturer’s instructions. Free-floating brain sections were permeabilized with 0.5% Triton X-100 and 3 mM ethylenediamine tetraacetic acid (EDTA) in 0.1 M PBS for 30 min. ProteoStat dye was then added at a 1:2000 dilution, and Hoechst dye was added at a 1:1000 dilution for 30 min at room temperature. Images were acquired with an FV3000 confocal laser microscope (Olympus).

### 2.6. Immunohistochemistry

After washing, free-floating brain sections were blocked with 5% donkey serum for 1 h and then treated with primary antibody against anti-iba-1 (Wako, Osaka, Japan) overnight at 4 °C. After three washes, the sections were incubated with an avidin-biotin complex solution (Vector Laboratories, Burlingame, CA, USA) for 1 h. After washing, the sections were developed with 3,3′-diaminobenzidine (Sigma–Aldrich, St. Louis, MO, USA) containing 0.025% H_2_O_2_. Sections were then dehydrated in graded alcohol solutions followed by xylene and mounted under coverslips with Permount (Sigma–Aldrich). A BX51 light microscope (Olympus) was used to acquire images of the stained sections.

### 2.7. Sholl Analysis of Microglial Morphology

ImageJ software (Version 1.52a, NIH, Bethesda, MD, USA) and the Sholl analysis plugin were used for manual morphological analysis for iba-1-positive microglia. As previously described [17], the maximum radius of the cell soma and the radius exceeding the cell’s most extended branch were measured and the number of primary branches was manually counted. The ramification index was calculated for each cell. It was calculated as the number of end branches divided by the number of primary branches. A minimum of 15–40 cells per imaging region were analyzed for each mouse.

### 2.8. Immunofluorescence

Free-floating brain sections or BV2 cells were treated with primary antibodies against synaptophysin (Santa Cruz Biotechnology, Dallas, TX, USA) and TonEBP (Santa Cruz) overnight at 4 °C, respectively. After washing, brain sections or cells were incubated with donkey secondary antibodies conjugated with Alexa Fluor 594 (Invitrogen Life Technologies, Carlsbad, CA, USA) for 1 h at RT. Then, 4′,6-diamidino-2-phenylindole (DAPI, Thermo Fisher Scientific, Waltham, MA, USA) was used to stain nuclei for 30 min. Fluorescence images were obtained using a BX51-DSU microscope (Olympus). The fluorescence intensity in each selected region was measured using i-Solution (IMT i-Solution Inc., Vancouver, BC, Canada).

### 2.9. Transmission Electron Microscopy (TEM)

After anesthesia, mice (*n* = 3) were perfused with 0.1 M PBS followed by 4% paraformaldehyde and 0.05% glutaraldehyde in PBS. Brains were post-fixed with the same fixative solution for 2 h at 4 °C and then cut into 60 μm coronal sections using a vibratome (Leica). For thin TEM sections, we prepared them as previously described [18]. Thirty-five micrographs (17.08 μm^2^) from each mouse were obtained and used for quantification. To determine the number of microglial contacts and perimeter of microglial contact, images were collected from the pyramidal cell layer of the CA1 region. Morphological properties were quantified, and measurements were made using ImageJ software (version 1.52a).

### 2.10. Western Blot Analysis

Brains were rapidly removed from mouse skulls, and both hippocampi were dissected and frozen. Hippocampal or BV2 cell lysates were homogenized individually in sterile tubes containing 150 mL tissue protein extraction reagent lysis buffer (Thermo Fisher Scientific). Homogenized tissues or cells were incubated on ice for 30 min and sonicated three times for 1 min. After centrifuging the supernatant of the samples were transferred to clean vials and kept at −80 °C. The total lysis extract (*n* = 3–7) was separated by sodium dodecyl sulfate (SDS)-polyacrylamide gel electrophoresis and the proteins were then transferred onto polyvinylidene difluoride (PVDF) membranes. Proteins were immunoblotted with cyclooxygenase-2 (COX-2, Santa Cruz), iba-1 (Wako), TonEBP (Santa Cruz), and TREM2 (Santa Cruz) antibodies. β-actin (Sigma Aldrich, St. Louis, MO, USA) was used as a control to normalize total protein levels. Protein bands were detected using an enhanced chemiluminescence substrate (Pierce, Rockford, IL, USA). The band density was analyzed using the Multi-Gauge V 3.0 (Fujifilm, Tokyo, Japan) image analysis application.

### 2.11. Quantitative Real-Time Reverse-Transcription PCR (qRT-PCR)

For telomere length analysis, the DNeasy Blood & Tissue Kit (Qiagen, Hilden, Germany) or the Wizard SV 96 Genomic DNA Purification System (Promega, Madison, WI, USA) were used to extract genomic DNA from brain tissues according to the manufacturers’ protocols. Telomere (Forward 5′-ACACTAAGGTTTGGGTTTGGGTTTGGGTTTGGGTTAGTGT-3′and Reverse 5′-TGTTAGGTATCCCTATCCCTATCCCTATCCCTATCCCTAACA-3′) 36b4 (Forward 5′-ACTGGTCTAGGACCCGAGAAG-3′ and Reverse 5′-TCAATGGTGCCTCTGGAGATT-3′) primers were used for qPCR. Sense and antisense primers were included in the optimized PCR mixture (10 μL, 384-well plate, triplicate analyses per sample, 900 nM each for telomeres, 500 nM each for 36b4). A Light Cycler 480 II was used for qPCR (Roche Molecular Diagnostics, Pleasanton, CA, USA). The optimized thermal program consisted of one cycle at 95 °C for 10 min, two cycles at 95 °C for 15 s, and one cycle at 49 °C for 15 s for pre-amplification. Melting curve analysis verified the homogeneity of the PCR products after qPCR. Telomere lengths were estimated in relation to the reference single-copy nuclear gene using the 2^−ΔCt^ equation: 2(^Cttar−Ct36b4^) (Ct, threshold cycle; Ct_tar_, Ct value of the target, telomere, Ct value of the reference single-copy gene, 36b4).

Total RNA was extracted from hippocampi and BV2 cells using TRIzol reagent (Invitrogen) and reverse-transcribed using the RevertAid™ First-Strand cDNA Synthesis Kit (Fermentas Inc., Hanover, MD, USA) according to the manufacturer’s protocol. qRT-PCR was performed using the ABI Prism 7000 Sequence Detection System (Applied Biosystems, Foster City, CA, USA). PCR amplifications were performed using a SYBR Green I qPCR Kit (TaKaRa, Shiga, Japan) with specific primers; tumor necrosis factor (*tnf*)-*α* (Forward 5′ CCAGACCCTCACACTCAGATC 3′ and Reverse 5′ CACTTGGTGGTTTGCTACGAC 3′), interleukin (*il*)*-6* (Forward 5′ AGTTGCCTTCTTGGGACTGA 3′ and Reverse 5′ TCCACGATTTCCCAGAGAAC 3′), inducible nitric oxide synthase (*inos*, Forward 5′ GGAGTGACGGCAAACATGACT 3′ and Reverse 5′ TCGATGCACAACTGGGTGAAC 3′), cyclooxygenase-2 (*Cox-2*, Forward 5′ AGGACTGGGCCATGGAGT 3′ and Reverse 5′ ACCTCTCCACCAATGACCTG 3′), arginase-1 (*arg1*, Forward 5′ AAAGCTGGTCTGCTGGAAAA 3′ and Reverse 5′ ACAGACCGTGGGTTCTTCAC 3′), transforming growth factor-*β1* (*Tgf-β1*, Forward 5′ TGGAGCAACATGTGGAACTC 3′and Reverse 5′ CAGCAGCCGGTTACCAAG 3′), *Il-10* (Forward 5′ CCAGGGAGATCCTTTGATGA 3′ and Reverse 5′ AACTGGCCACAGTTTTCAGG 3′), and *Gapdh* (Forward 5′ AAATGGTGAAGGTCGGTGTG 3′and Reverse 5′ CATGTAGTTGAGGTCAATGAAGG 3′). Relative quantification was performed using the ∆∆Ct formula, and relative mRNA gene expression levels were expressed as fold changes relative to a calibrator sample.

### 2.12. Microglial BV2 Cell Culture

The murine microglial BV2 cell line was acquired from the Korean Cell Line Bank (Seoul, Republic of Korea). Under normal culture conditions, cells were grown in Dulbecco’s Modified Eagle’s Medium (DMEM; Gibco, Grand Island, NY, USA) supplemented with 10% fetal bovine serum (FBS; Gibco), 100 U/mL penicillin, and 100 U/mL streptomycin (pen/strep; Gibco) at 37 °C in a 5% CO_2_ humidified atmosphere. Cells were detached in 0.05% Trypsin-EDTA (Gibco) and plated at a density of 3 × 10^5^ cells per 60 mm dish. Cells were treated with 100 µM AβO for a series of concentrations or 24 h.

### 2.13. siRNA Transfection

Cells were transfected with TonEBP small interfering RNA (siRNA) or control scrambled siRNA (Bioneer Corp., Daejeon, Republic of Korea) using lipofectamine RNAiMAX reagent (Invitrogen Life Technologies) following the manufacturer’s instructions.

### 2.14. Wound-Induced Migration Assay

BV2 cells were plated in 6-well plates at a density of 3 × 10^5^ cells/well and incubated at 37 °C until reaching 80–90% confluence after TonEBP siRNA transfection. When confluency was reached, cells were incubated with serum-starved medium for 12 h and treated with 2 µM AβO for 16 h after wounding with a tip. The scratched area was monitored with an Olympus IX71 phase-contrast inverted microscope, and images were obtained at 0 and 16 h. The wound area was calculated by ImageJ (version 1.52a). The degree of cell migration is presented as a percentage of vehicle-treated control cells.

### 2.15. Transwell Migration Assay

To assess microglial migration, TonEBP siRNA-transfected BV2 cells (1 × 10^5^) were seeded in the top chamber of a Transwell plate (8 µm pore size, Merck Millipore, Billerica, MA, USA) with serum-free DMEM, and the lower chamber was filled with medium containing 10% FBS. After cells were attached for 12 h, they were treated with or without Aβ and cultured for 24 h at 37 °C. The top chamber was then removed, and cells that migrated to the lower chamber were fixed with 4% paraformaldehyde for 20 min and stained with Cresyl violet for 20 min. Following PBS washing, stained cells were visualized with a phase-contrast inverted microscope (Olympus).

### 2.16. Phagocytosis Assay

To determine the phagocytic function of AβO or siTonEBP, BV2 cells were incubated with 20 μg/mL pHrodo^TM^ Red Zymosan A Bioparticles (Invitrogen Life Technologies) at 37 °C for 1 h. Nuclei were stained with DAPI (1:20,000, Thermo Fisher Scientific) following fixation with 4% paraformaldehyde for 15 min. Fluorescent slides were visualized using an FV3000 confocal laser microscope (Olympus).

### 2.17. Behavioral Tests

To assess short-term spatial memory, we tested mice in the Y-maze, which consisted of three arms (44 cm × 10 cm × 15 cm) that met at an angle of 120°. Each mouse was allowed to freely explore arm A for 5 min. After 1 h, the mouse was returned to arm A with their head pointed away from the maze’s center and permitted to explore the entire maze, including arms B and C for 15 min. The number of entries into each arm was recorded, and the percentage of spontaneous alternations was calculated by dividing the number of successful alternations (e.g., ABC and BCA but not CCA) by the total number of alternation triads.

To assess long-term memory, we tested mice in the novel object recognition (NOR) test as previously described [19] with the following modifications. On the first day, mice (*n* = 4–6) were brought to the testing room 20 min before the experiment to become acquainted with the surroundings. Each mouse was then permitted to freely explore an open-field box (25 cm × 25 cm × 25 cm) for a 5 min habituation session. During the training session, two identical objects were placed in opposing locations within the box, each at the same distance from the nearest corner, and the mouse was given 10 min to investigate the objects before being returned to its home cage. After 1 h, the mouse was returned to the box, in which one of the two familiar objects was replaced with a new object for a 5 min trial (test session). All objects were different in form and color but similar in size and were secured to the box’s floor to prevent movement. The entire box and objects were cleaned with water after each session to eliminate the possibility of odor signals. Object exploration time was defined as the amount of time a mouse spent with its nose within 2 cm of an object while smelling or pawing the object. Sitting or standing on an object was not considered exploration. Exploration time was carefully measured using two stopwatches. During the training session, the object bias was calculated as (time exploring object A ÷ total time exploring objects A and B) × 100%. During the testing session, novelty preference was calculated as (time exploring novel object ÷ time exploring familiar and novel objects) × 100, and discrimination index was calculated as (time exploring novel object—time exploring familiar object ÷ time exploring novel and familiar objects).

To assess spatial memory, mice were tested in a Morris water maze (MWM) as previously described [20] with the following modifications. Briefly, mice (*n* = 5–6) were trained to find a randomly positioned visible platform in a 120 cm diameter swimming pool filled with water maintained at 24 ± 1 °C. All mice received four trials per day for four consecutive days, and the starting position was changed with each trial. Escape latency was recorded by a video-tracking program (Noldus EthoVision XT7, Noldus Information Technology, Wageningen, The Netherlands). On the day of testing, the platform was removed, and the number of crossings over the target zone (i.e., where the platform was located during training) was measured.

### 2.18. Statistical Analysis

All statistical analyses were performed using PRISM 7.0 (GraphPad Software Inc., San Diego, CA, USA). To determine group differences, we used Student *t*-tests and one-way analysis of variance (ANOVA), followed by Tukey post hoc analyses. Results are expressed as the mean and standard error of the mean (SEM). A *p*-value < 0.05 was considered statistically significant.

## 3. Results

### 3.1. Hippocampal TonEBP Expression Increases in Aged Mice with Microglial Activation

Given that TonEBP is implicated in the regulation of neuroinflammation [11,14], we investigated whether the hippocampal expression of TonEBP changes with age. As expected, we found that both middle-aged (12-month-old) and old (24-month-old) mice exhibited higher levels of hippocampal TonEBP expression than young (6-month-old) mice (Figure 1a). Neuroinflammation is associated with activated microglia in the aged brain [21]. Using ProteoStat aggresome dye, we found increased protein aggresome accumulation within the hippocampus with age (Figure 1b). In accordance with the increases in aggresome level, the expression levels of hippocampal triggering receptor expressed on myeloid cells 2 (TREM2) and Iba-1, which are protein markers of microglial activation, were increased in middle-aged and old mice compared with young mice (Figure 1c). When microglia are fully activated, they take on an amoeboid shape and begin to participate in phagocytosis [22]. Therefore, Sholl analysis was performed to quantify microglial activation and revealed that the numbers of intersections and the ramification index of microglial processes within the hippocampal CA1 region were reduced in middle-aged and old mice compared with those in young mice (Figure 1d–f). These findings indicate that TonEBP may contribute to age-related microglial activation in the mouse hippocampus.

### 3.2. TonEBP Haploinsufficiency Attenuates Microglial Activation in the Hippocampi of Middle-Aged Mice

Given the association of increased TonEBP expression with microglial activation, to examine whether TonEBP haploinsufficiency delays aging progression, we measured telomere length in hippocampal tissues from young and middle-aged mice. Telomere length was significantly reduced in middle-aged WT mice compared with young WT mice (Figure 2a). However, this reduction in telomere length in middle-aged mice was inhibited by TonEBP haploinsufficiency. As expected, hippocampal TonEBP expression was decreased in middle-aged TonEBP (+/−) mice compared with WT mice (Figure 2b). In addition to a lower COX-2 protein level, the expression levels of hippocampal TREM2 and Iba-1 were also reduced in TonEBP (+/−) mice (Figure 2b). Furthermore, we found that the expression level of hippocampal *Tnf-a* mRNA was significantly reduced in TonEBP (+/−) mice compared to WT mice, but not *il-6* mRNA (Figure 2c). Sholl analysis of microglial morphology revealed that middle-aged WT mice had activated microglia, but TonEBP (+/−) mice had ramified microglia (Figure 2d–f). These findings suggest that TonEBP may be crucial for neuroinflammation with microglial activation in the middle-aged brain.

### 3.3. TonEBP Haploinsufficiency Attenuates Cognitive Impairment in Middle-Aged Mice

We next investigated the effect of TonEBP haploinsufficiency on cognitive impairment using the Y-maze, NOR, and MWM tests (Figure 3). In the Y-maze, TonEBP (+/−) mice showed a higher rate of alternation than WT mice, indicating that TonEBP haploinsufficiency attenuated memory deficits in middle-aged mice (Figure 3a). In the NOR test, TonEBP (+/−) mice spent more time exploring a novel object than WT mice and also exhibited a higher novelty preference and discrimination index (Figure 3b). In the MWM test, TonEBP (+/−) mice exhibited shorter escape latencies during training than WT mice and more crossings over the target zone during the test (Figure 3c). These findings indicate that TonEBP can be associated with cognitive impairment in middle-aged mice.

### 3.4. TonEBP Haploinsufficiency Inhibits Atypical Dendritic Spines and Synaptic Stripping in the Hippocampi of Middle-Aged Mice

The loss of dendritic spines is closely linked to memory deficits in the old brain [23]. To determine whether TonEBP haploinsufficiency inhibits dendritic spine loss in middle-aged mice, Golgi staining was performed to evaluate changes in basal and apical dendritic spines in the hippocampal CA1 region (Figure 4a). TonEBP (+/−) mice exhibited increases in the proportions of basal and apical dendrite spines with a mushroom shape compared with WT mice (Figure 4b–f). To evaluate the synaptic density, we performed immunofluorescence for synaptophysin, which is a major synaptic vesicle membrane protein (Figure 4g). TonEBP (+/−) mice showed greater synaptophysin-stained puncta of neurons compared with WT mice. Moreover, microglial phagocytosis is known to contribute to synaptic stripping [24], with microglial processes frequently containing cellular inclusions resembling dendritic spines or axon terminals, suggesting their phagocytic engulfment. Therefore, to evaluate whether TonEBP haploinsufficiency attenuates synaptic stripping, we performed TEM analysis and found that TonEBP haploinsufficiency reduced the number and perimeter of microglia in contact with dendritic spines in the hippocampi of middle-aged mice (Figure 4h,i). These findings indicate that TonEBP haploinsufficiency protects against age-induced synaptic damage.

### 3.5. TonEBP Haploinsufficiency Attenuates Microglial Activation and Cognitive Impairment in the Hippocampi of AβO-Injected Mice

Given that Aβ is a pathological hallmark of AD, we then used an AβO-injected AD-like cognitive deficits mouse model to evaluate the effects of TonEBP haploinsufficiency [25]. In line with lower hippocampal TonEBP expression in AβO-injected TonEBP (+/−) mice, the expression levels of AβO-induced COX-2, TREM2, and Iba-1 proteins were reduced by TonEBP haploinsufficiency (Figure 5a). Mice were then subjected to Y-maze tests to determine whether TonEBP haploinsufficiency improves AβO-induced memory loss. AβO-treated TonEBP (+/−) mice showed a higher rate of alteration compared with AβO-treated WT mice (Figure 5b). Golgi staining also revealed that TonEBP (+/−) mice had more dendritic spines than WT mice (Figure 5c,d). In particular, TonEBP (+/−) mice showed increases in the proportions of basal and apical dendrite spines with a mushroom shape compared with WT mice (Figure 5e,f). These findings indicate that TonEBP haploinsufficiency reduces AβO-induced microglial activation, synaptic loss, and memory deficits.

### 3.6. TonEBP Knockdown Reduces AβO-Induced Inflammation in BV2 Cells

A previous study reported that TonEBP enhances the expression of proinflammatory genes induced by LPS treatment in BV2 cells [13]. To examine the effect of TonEBP inhibition on microglial activation, we examined AβO-treated BV2 cells in vitro. As expected, TonEBP was elevated in AβO-treated BV2 cells in a dose-dependent manner (Figure 6a). TonEBP siRNA significantly inhibited AβO-induced TonEBP expression (Figure 6b,c). Furthermore, TonEBP siRNA decreased proinflammatory *inos* and *Cox-2* mRNA expression and increased anti-inflammatory *Arg1*, *Il-10*, and *Tgf-β1* mRNA expression levels (Figure 6d,e).

### 3.7. TonEBP Knockdown Reduces AβO-Induced Microglial Migration and Phagocytosis in BV2 Cells

In addition to the generation of proinflammatory cytokines, the migration of microglial cells toward sites of injury in response to inflammatory stimuli also plays a crucial role in the spread of neuroinflammation [26]. To verify whether TonEBP inhibition impacts AβO-induced migration of BV2 cells, we performed wound healing and Transwell assays (Figure 7a,b). In both assays, siRNA-mediated TonEBP knockdown inhibited AβO-induced migration of BV2 cells. After macrophages are exposed to inflammatory stimuli, phagocytic receptors on macrophages bind zymosan and stimulate particle engulfment [27]. Therefore, we measured phagocytosis using a zymosan assay kit. As shown in Figure 7c, TonEBP knockdown significantly inhibited the increase in zymosan particles in AβO-treated BV2 cells. These findings suggest that TonEBP inhibition reduces AβO-induced inflammation and leads to less microglial migration and phagocytosis in BV2 cells.

## 4. Discussion

The present study made four major observations: (i) mouse hippocampal TonEBP increases with aging; (ii) TonEBP haploinsufficiency reduces microglial activation, decreases the number and perimeter of microglia contacting dendritic spines, and improves memory deficits in middle-aged mice; (iii) TonEBP haploinsufficiency reduces microglial activation and synaptic loss in AβO-treated mice; (iv) TonEBP deletion reduces microglial migration and phagocytosis in AβO-treated BV2 cells. For the first time, we report that TonEBP increases with aging. Based on these findings, we propose that TonEBP plays an important role in microglial activation and synaptic degeneration in the aged and AβO-treated brain.

Our previous studies implicated TonEBP in neuroinflammation in mouse models of seizure and diabetes [10,12]. Other recent studies demonstrated that myeloid-specific TonEBP deletion protects against LPS-induced neuroinflammation and memory loss [13]. Consistent with the induction of TonEBP-positive microglia in LPS-treated primary microglia or rats with middle cerebral artery occlusion [11], we demonstrated that TonEBP expression increased in a dose-dependent manner after AβO injection into the mouse hippocampus and that its expression was blocked by TonEBP siRNA in AβO-treated BV2 cells. These findings suggest that microglial TonEBP plays an important role in the progression of aging and AD.

Jeong et al. showed that TonEBP deficiency attenuates proinflammatory cytokines such as iNOS and COX-2 in LPS-treated BV2 cells [13]. Furthermore, they showed that myeloid-specific TonEBP deletion reduces LPS-induced microglial activation and memory loss. Notably, we found that TonEBP knockdown decreased inflammatory *inos* and *Cox-2* mRNA expression and increased anti-inflammatory *Arg1*, *Il-10*, and *Tgf-ß1* mRNA expression in AβO-treated BV2 cells. Consistent with our findings, previous studies have shown that AβO treatment induces microglial phagocytosis and neuroinflammation in BV2 cells [28,29]. It has also been demonstrated to mediate the effects of AβO in BV2 cells during a phagocytic response [4]. In this study, along with the inhibition of AβO-induced migration, a zymosan assay revealed that siTonEBP significantly attenuated increased zymosan particles in AβO-treated BV2 cells. These results suggest that inhibiting TonEBP-mediated inflammation protects against AβO-induced microglial activation and leads to less microglial migration and phagocytosis in BV2 cells.

Some studies have also reported that COX-2 expression is closely associated with inflammation and neurodegeneration in Parkinson’s disease and AD [30,31]. The long-term administration of non-steroidal anti-inflammatory drugs reduces the risk of AD by inhibiting the expression of COX-2-mediated reciprocal management of IL-1β and Aβ between glial cells and neurons [32,33]. In the present study, we confirmed that the increased levels of hippocampal COX-2 protein in aged mice were reduced in AβO-treated TonEBP (+/−) mice. Thus, we suggest that TonEBP promotes neuroinflammation in the aged brain.

TREM2, which is primarily produced by microglia, is implicated in the phagocytosis of apoptotic neurons, damaged synapses, and amyloid plaques [34,35,36,37]. In particular, TREM2 accumulation is closely associated with microglial phagocytosis, as TREM2 was found to enhance the Aβ phagocytic activity of microglia via activation of C/EBPα-dependent CD36 expression [38]. Conversely, TREM2 deletion impairs Aβ phagocytosis by reducing microglial migration [36], and mutation of TREM2 results in defective phagocytosis and the development of AD [39], supporting other findings that microglial activation is accompanied by increased TREM2 during the progression of AD [40]. However, another study showed that reduced TREM2 expression inhibits proinflammatory cytokines in LPS-treated microglia by inhibiting NF-κB signaling [41]. Based on this contradictory evidence, TREM2 appears to exert anti-inflammatory or proinflammatory effects depending on its stimulus or aging-related neurodegenerative disease. In the present study, we demonstrated that TREM2 increased with aging and that TonEBP haploinsufficient mice showed lower hippocampal TREM2 expression than WT mice at 12 months of age. TREM2 was also reduced in the AβO-injected hippocampi of TonEBP haploinsufficient mice compared with WT mice. In addition, AβO-induced migration of BV2 cells was decreased by TonEBP knockdown. Taken together, we hypothesize that inhibiting TREM2 via TonEBP haploinsufficiency contributes to the regulation of microglial activation and phagocytosis in the hippocampi of aged or AβO-treated mice.

Previous studies have reported that microglia function in synaptic pruning, maintaining neuronal survival, axonal projection sculpting, and synaptic plasticity during learning in the adult brain [42,43,44,45]. In particular, aging-induced microglial activation is associated with neuroinflammation, dendritic spine loss, and cognitive impairment [45,46]. Although microglia activated by Aβ produce proinflammatory cytokines and exacerbate neuronal damage, microglia also play a vital role in synaptic pruning, altering the balance between excitatory and inhibitory neuronal activity [7]. For example, sustained LPS-induced microglial activation is associated with the synaptic stripping of inhibitory terminals in the motor cortex [43]. In the present study, we found increased basal and apical synaptic densities in the hippocampi of aged or AβO-treated TonEBP (+/−) mice compared with WT mice. In addition, TonEBP (+/−) mice had a larger number of dendritic spines, particularly those that were mushroom-shaped. Interestingly, TEM analysis revealed that TonEBP haploinsufficiency reduced the number of microglia contacting dendritic spines in the aged brain. These results are consistent with previous findings that microglia–synapse interactions are regulated by neuronal activity [47]. Microglia continuously monitor the neuropil with motile processes for the clearance of cellular debris and damaged neurites [7]. Hao et al. reported that memory deficits in obese mice are attributable to synaptic stripping by microglia [48]. Thus, our findings suggest that TonEBP enables microglia to respond to altered synaptic morphology during the progression of aging.

Our study has several limitations. First, we did not use aged 3xTg-AD mice with Aβ deposition; therefore, the precise role of TonEBP during aging was not evaluated, which may limit the generalizability of our findings. Second, the direct interaction between neurons and microglia mediated by TonEBP was not assessed. Third, we only investigated microglia in the hippocampi of middle-aged mice, although TonEBP has been shown to play critical roles in several neurodegenerative diseases.

## 5. Conclusions

This study demonstrates that TonEBP haploinsufficiency inhibited aging-related microglial activation and synaptic pruning and attenuated cognitive impairment. Furthermore, TonEBP deletion reduced microglial migration and phagocytosis in AβO-treated BV2 cells. Thus, targeting TonEBP may serve as an approach for reducing neuroinflammation and restoring brain health during the course of aging and in cases of neurodegeneration to improve cognitive function.

## Figures and Tables

**Figure 1 cells-12-02612-f001:**
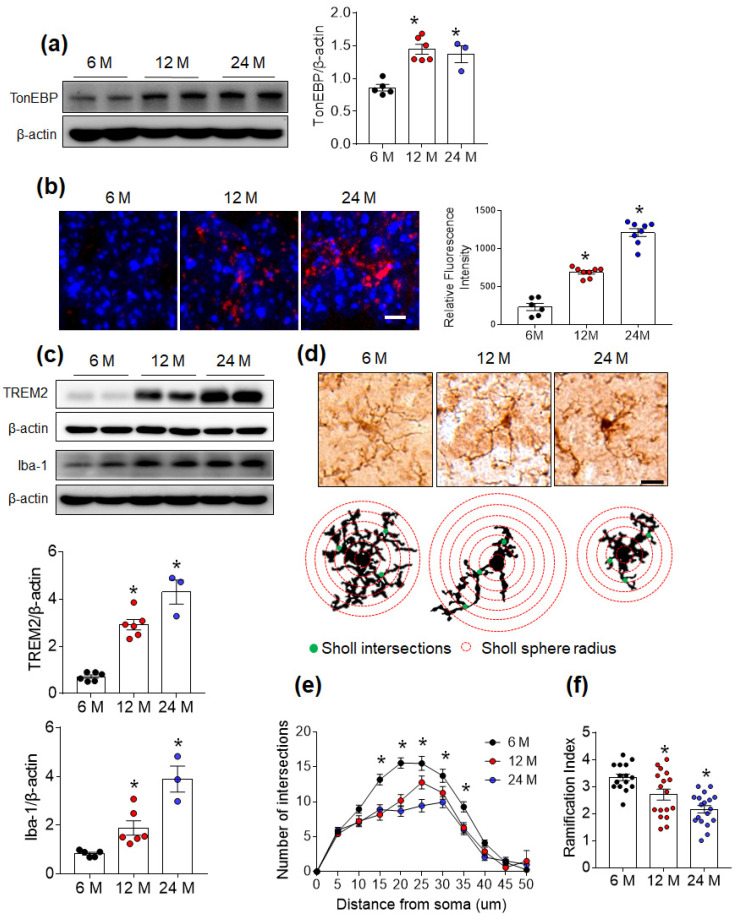
TonEBP and microglial activation increase with age in the mouse hippocampus. (**a**) Western blot and quantitative analysis of TonEBP protein level in the hippocampi of 6-, 12-, and 24-month-old mice (*n* = 3–7). (**b**) Representative images of the ProteoStat staining assay in the CA1 region of hippocampal sections. The intensity of the ProteoStat dye is presented as fold changes. DAPI was used to stain nuclei. Scale bar = 10 µm. (**c**) Western blot and quantitative analysis of TREM2 and Iba-1 protein expression in the hippocampus (*n* = 3–7). (**d**) Representative microglial morphology and schemes graphically illustrate the calculation bases for the intersections (green spots) and ramification indices (red circles) for microglial activation. Scale bar = 10 μm. (**e**,**f**) Sholl analysis of intersections (**e**) and ramification indices (**f**). Data are presented as the mean ± SEM. * *p* < 0.05 vs. 6-month-old mice.

**Figure 2 cells-12-02612-f002:**
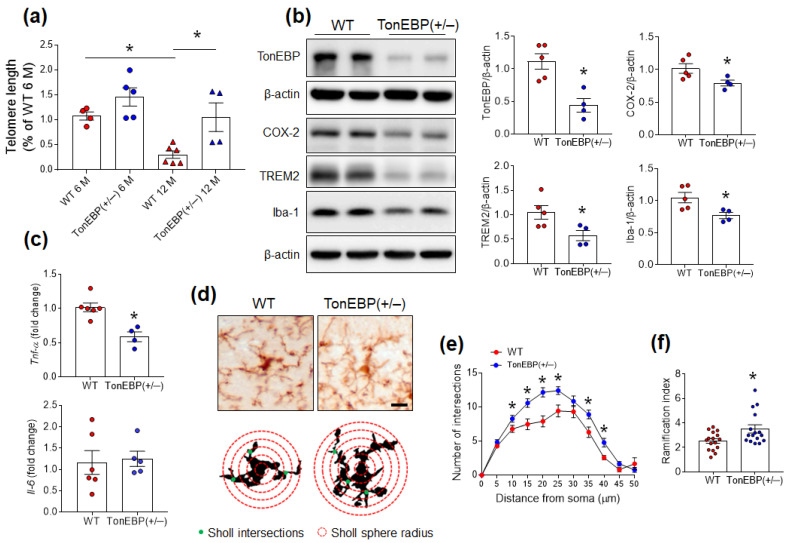
TonEBP haploinsufficiency attenuates microglial activation in the hippocampi of middle-aged mice. (**a**) Relative telomere length in the hippocampi of 6- and 12-month-old WT and TonEBP (+/−) mice. Telomere length (*n* = 4–6) was measured by qRT-PCR. (**b**) Western blot and quantitative analysis of TonEBP, COX-2, TREM2, and Iba-1 protein expression in the hippocampi of 12-month-old WT and TonEBP (+/−) mice (*n* = 4–5). (**c**) Effects of TonEBP haploinsufficiency on *Tnf-α* and *Il-6* mRNA expression using qRT-PCR. (**d**) Representative microglial morphology and schemes graphically illustrate the calculation bases for the intersections (green spots) and ramification indices (red circles) for microglial activation. Scale bar = 10 μm. (**e**,**f**) Sholl analysis of intersections (**e**) and ramification indices (**f**). Data are presented as the mean ± SEM. * *p* < 0.05 vs. WT.

**Figure 3 cells-12-02612-f003:**
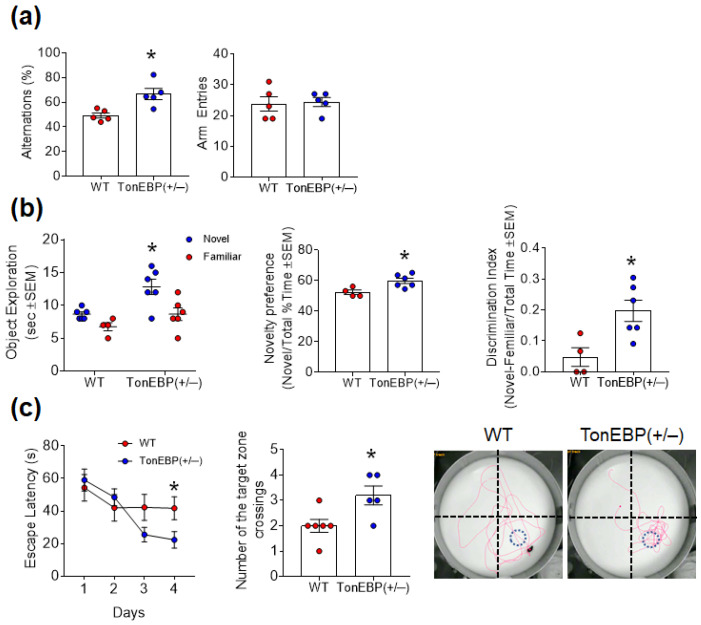
TonEBP haploinsufficiency inhibits cognitive impairment in middle-aged mice. (**a**) Percentage of spontaneous alternations and total arm entries in the Y-maze test. (**b**) Performance in the novel object recognition test in terms of raw exploration time, novelty preference, and discrimination index. (**c**) Escape latency, number of target zone crossings, and representative swim paths in the Morris water maze. Red line and blue circle are swim path and the platform, respectively. Data are presented as the mean ± SEM. * *p* < 0.05 vs. WT.

**Figure 4 cells-12-02612-f004:**
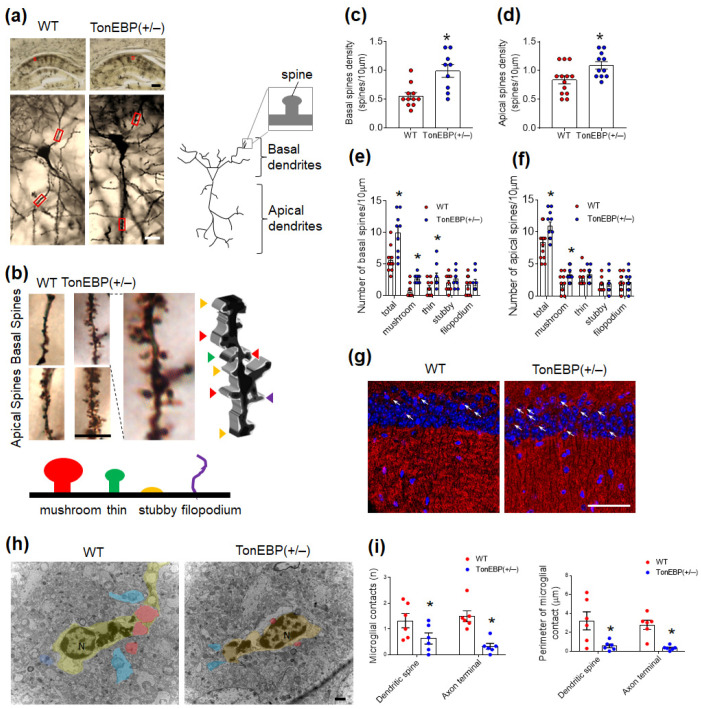
TonEBP haploinsufficiency reduces atypical dendritic spines and synaptic stripping in the hippocampi of middle-aged mice. (**a**) Representative microphotographs of Golgi-stained hippocampi from WT and TonEBP (+/−) mice. Upper and lower red box are basal and apical dendrite, respectively. Scale bars = 500 μm (top), 20 μm (bottom). (**b**) Representative, higher-resolution microphotographs of Golgi-stained neurites in the hippocampi of WT and TonEBP (+/−) mice. Different arrowhead colors indicate each different spine shape. Spine maturity progresses from mushroom spines (red) to thin (green), stubby (yellow), and filopodial (purple) spines. Scale bar = 10 μm. (**c**,**d**) Analysis of the basal (**c**) and apical (**d**) spine densities. (**e**,**f**) Analysis of the numbers of basal (**e**) and apical (**f**) spines according to spine shape. (**g**) Representative immunofluorescence for synaptophysin in the CA1 region of hippocampi. Scale bar = 10 μm. DAPI was used to stain nuclei. White arrows indicate synaptic vesicle of neuron. (**h**) Representative electron microscopy images in hippocampal regions showing multiple cellular inclusions in microglia. A microglial process (green) contacts several axon terminals (blue) and dendritic spines (red) along with the extracellular space area. Scale bar = 0.5 μm. (**i**) Total numbers of contacts and average perimeters of contacts between microglial processes and synapse-associated elements (i.e., dendritic spines and axon terminals). *n* = 40 microglial processes per group (*n* = 3). Data are presented as the mean ± SEM. * *p* < 0.05 vs. WT.

**Figure 5 cells-12-02612-f005:**
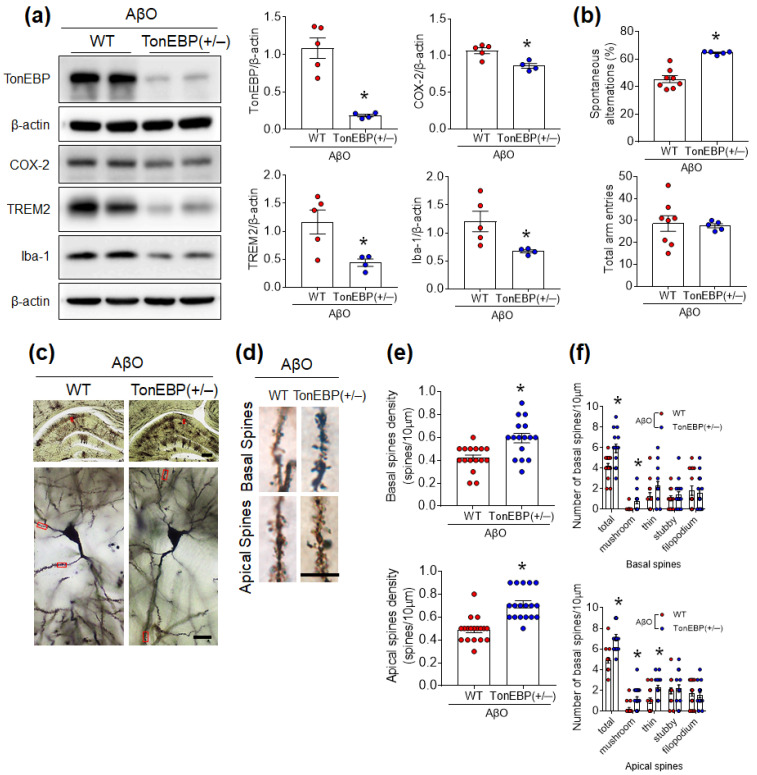
TonEBP haploinsufficiency inhibits neuroinflammation and cognitive decline in the hippocampi of AβO-treated mice. (**a**) Western blot and quantitative analysis of TonEBP, COX-2, TREM2, and Iba-1 expression in the hippocampi of AβO-injected WT and TonEBP (+/−) mice (*n* = 4–5). (**b**) Percentage of spontaneous alternations and total arm entries in the Y-maze. (**c**) Representative microphotographs of Golgi-stained hippocampi. Scale bar = 500 μm (top), 20 μm (bottom). (**d**) Representative, higher-resolution microphotographs of Golgi-stained neurons (red boxes in **c**). Scale bar = 10 μm. (**e**) Analysis of the basal and apical spine densities. (**f**) Analysis of the numbers of basal and apical spines according to spine shape. Data are presented as mean ± SEM. * *p* < 0.05 vs. AβO-treated WT mice.

**Figure 6 cells-12-02612-f006:**
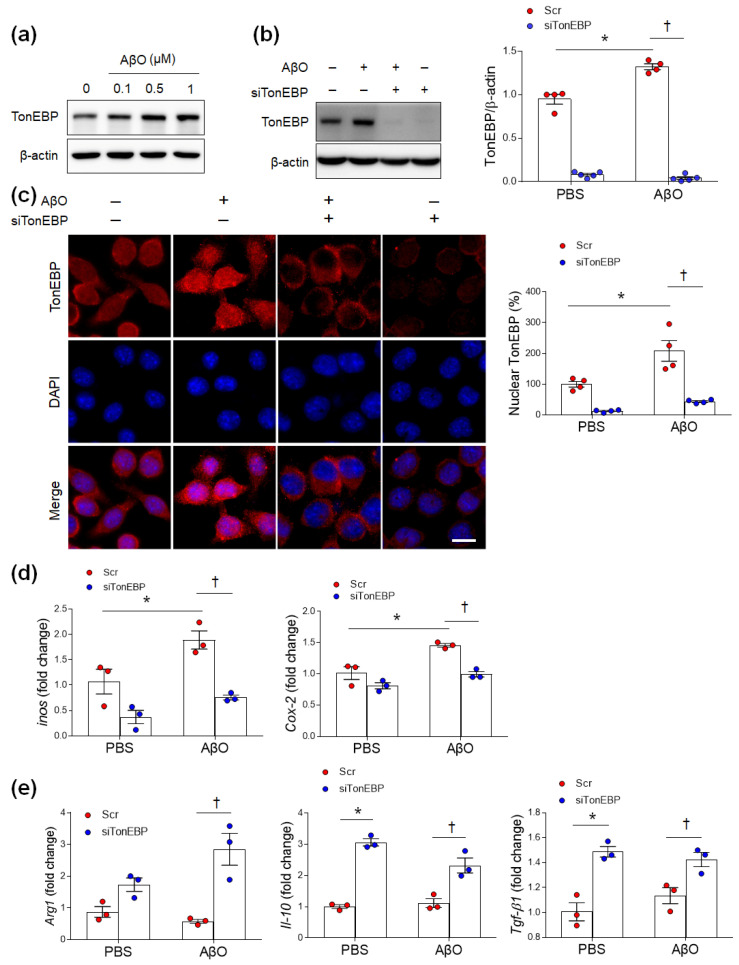
TonEBP knockdown reduces AβO-induced inflammation and migration in BV2 cells. (**a**) Western blot analysis of TonEBP protein expression in BV2 cells. The cells were treated for 15 h with 0.1, 0.5, or 1 μM AβO. (**b**) Cells were transfected with scrambled (Scr) or 20 nM TonEBP-targeted siRNA (siTonEBP) followed by a 15 h treatment with 2 μM AβO. (**c**) Representative micrographs and quantification of TonEBP in BV2 cells. Scale bar = 10 μm. DAPI was used to stain nuclei. Data are presented as the mean ± SEM * *p* < 0.05 vs. control (PBS, Scr). (**d**,**e**) Effects of TonEBP knockdown on the mRNA expression levels of M1 (*inos* and *Cox-2*) and M2 (*Arg1*, *IL-10*, and *TGF-β1*) genes in AβO-treated BV2 cells using qRT-PCR. Data are presented as the mean ± SEM from three independent experiments. * *p* < 0.05 vs. PBS/Scr; ^†^
*p* < 0.05 vs. AβO/Scr.

**Figure 7 cells-12-02612-f007:**
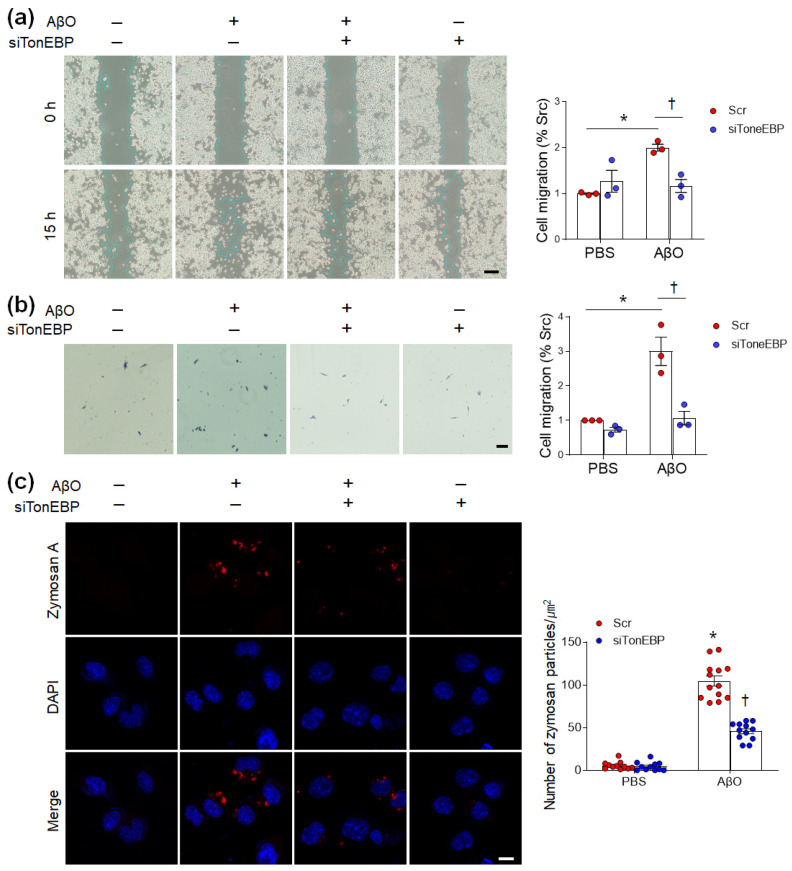
TonEBP knockdown reduces AβO-induced migration and phagocytosis in BV2 cells. (**a**,**b**) Cell migration was assessed by wound healing (**a**) and Transwell (**b**) migration assays. Data are presented as the mean ± SEM from three independent experiments. Scale bars = 20 μm (**a**) and 100 μm (**b**). (**c**) Representative images obtained after 1 h of incubation with zymosan red particles in AβO-treated BV2 cells. DAPI was used to stain nuclei. Bar graphs indicate the number of zymosan red particles. * *p* < 0.05 vs. PBS/Scr; ^†^
*p* < 0.05 vs. AβO/Scr. Scale bar = 10 μm.

## Data Availability

The data presented in this study are available on request from corresponding author.

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
