# Peer review of "TonEBP Haploinsufficiency Attenuates Microglial Activation and Memory Deficits in Middle-Aged and Amyloid β Oligomer-Treated Mice"

_cells, 2023, doi:10.3390/cells12222612_

Round 1
Reviewer 1 Report (Previous Reviewer 1)
Comments and Suggestions for Authors
Removing the data in Fig. 1 about the accumulation of Abeta in the wild type mice has greatly improved the manuscript quality and significance.
Reviewer 2 Report (Previous Reviewer 2)
Comments and Suggestions for Authors
None
This manuscript is a resubmission of an earlier submission. The following is a list of the peer review reports and author responses from that submission.
Round 1
Reviewer 1 Report
Comments and Suggestions for Authors
In this paper Lee et al. investigated the roles of TonEBP in microglial activation and cognitive impairment in an aged mouse model. They also evaluated the effects of TonEBP haploinsufficiency on neuroinflammation and dendritic spine loss in Aêžµ oligomer (AβO)–treated mice and the effects of TonEBP deletion on micro- 65 glial migration and phagocytosis on AβO-treated BV2 microglial cells. The study is of possible interest but there is major discrepancy that makes it not suitable for publication unless this issue is clarified. The problem regards Fig.1. The authors measured Ab levels and accumulation (Fig.1 b-c) in wildtype mice at different ages using Aβ1-16 antibody (6E10). It is established from several published papers that this specific antibody recognizes human APP and therefore human Aβ. It is also well known that endogenous Aβ does not aggregates and accumulates in the brain of wild type mice. How do the authors can explain these findings?
Reviewer 2 Report
Comments and Suggestions for Authors
Tonicity-responsive enhancer-binding protein (TonEBP) is a transcriptional factor in the Rel family which includes NF-κB and has previously been reported to mediate microglial activation in response to brain insults. Here, the authors investigated the role of TonEBP in the process of brain aging associated with memory deficits in TonEBP (+/-) and (+/+) mice and also after they received brain injections of amyloid êžµ oligomers (AêžµO). Hippocampal Aêžµ and TonEBP expression levels and microglial activation were all increased in 12 month and 24 month old (+/+) mice compared to 6 month old mice. 12 month old TonEBP (+/-) mice showed reduced levels of microglial activation and relatively preserved memory compared with 12 month old (+/+) mice. Electron microscopy revealed synaptic pruning by microglia processes in TonEBP (+/+) mouse brain which was much reduced in TonEBP (+/-) mice as was neuronal dendritic spine loss and memory deficits induced by brain AβO-innoculation. The effect of TonEBP knockdown on AêžµO-exposed BV2 microglial cells was also separately studied. It was found to attenuate microglial cell migration and phagocytosis. The authors suggest that TonEBP plays important roles in facilitating microglial activation both during aging and Alzheimer’s disease.
This paper reports a series of studies which extends their previous work on the role of TonEBP in epilepsy and stroke: I have some comments:
First, the paper implies that TonEBV plays a direct role in mediating microglial activation in aging but this is never directly shown here. While it is clear that TonEBV haploinsufficiency counteracts the rises in microglial activation and impaired recall in (+/+) mice associated with aging, or as a response to innoculated brain AêžµO, this could be directly mediated via a fall in NF-κB activity or some other mechanism in the TonEBP (+/-) mice.
Second, as it is loss of synaptic density that causes cognitive deficits in aging and Alzheimer’s disease, it is a little surprising that these workers did not directly measure synaptic density in their TonEBP (+/-) and (+/+) aging mice using synaptophysin staining or an SV2A protein marker. Instead, they have used EM to measure dendritic spines which is a rather indirect measure of synaptic loss.
Third, do these workers believe that TonEBP regulates expression of all microglial
phenotypes or just the phagocytic phenotype most influenced by TREM2 expression. Do the TonEBP (+/-) show reduced cytokine levels at 12 months compared to (+/+) mice.
In summary, this is an interesting study but the conclusion that TonEBP mediates microglial function in aging needs to be more cautious.
Round 2
Reviewer 1 Report
Comments and Suggestions for Authors
Even with the new data presented as Fig. 1b and 1c, the main problem remains. Abeta does not accumulate and form plaques in Nontg animals. Therefore this data is still not convincing.